# Substance Use among Spanish Adolescents: The Information Paradox

**DOI:** 10.3390/ijerph17020627

**Published:** 2020-01-18

**Authors:** Angel Belzunegui-Eraso, Inma Pastor-Gosálbez, Laia Raigal-Aran, Francesc Valls-Fonayet, Sonia Fernández-Aliseda, Teresa Torres-Coronas

**Affiliations:** 1Social Inclusion, Rovira i Virgili University, 43002 Tarragona, Spain; sonia.fernandeza@urv.cat (S.F.-A.); teresa.torres@urv.cat (T.T.-C.); 2Medical Anthropology Research Center, Rovira i Virgili University, 43002 Tarragona, Spain; inma.pastor@urv.cat; 3Department of Nursing, Rovira i Virgili University, 43002 Tarragona, Spain; laia.raigal@urv.cat (L.R.-A.); francesc.valls@urv.cat (F.V.-F.)

**Keywords:** adolescents, alcohol, tobacco, cannabis, information on drugs

## Abstract

This aim of this paper is to determine the relationship between the consumption of tobacco, cannabis, and alcohol (including drunkenness and binge drinking consumption patterns) in the previous 30 days by Spanish adolescents and the information that is available to adolescents on drug consumption. This cross-sectional study employed data from the Survey on Drug Use in Secondary Education in Spain (ESTUDES 2016), which was conducted on students aged 14 to 18 (*n* = 35,369). Contingency tables, mean comparison tests, and logistic regression analyses were conducted and prevalence ratios (PR) were obtained. The results show that the probability that an adolescent will smoke tobacco is associated with whether their mother and/or father smoke (PR: 1.30), whether some of their friends smoke (PR: 14.23), whether the majority of their friends smoke (PR: 94.05) and how well informed they perceive themselves to be (PR: 1.30). Cannabis use is mainly associated with whether most of their friends also use cannabis (PR: 93.05) and whether they are sufficiently informed regarding this consumption (PR: 1.59). Alcohol consumption is associated with whether their mothers drink regularly (PR: 1.21), whether most of their friends drink (PR: 37.29), and whether they are well informed (PR: 1.28). Getting drunk and binge drinking are associated with whether their friends have these behaviors (PR: 44.81 and 7.36, respectively) and whether they are sufficiently informed (PR: 1.23 for both behaviors). In conclusion, the consumption of these substances is more frequent among Spanish adolescents who believe that they are better informed and whose friends have similar patterns of consumption.

## 1. Introduction

The consumption of alcohol and other substances by adolescents is a public health issue in Spain. This is despite the fact that the ESTUDES survey (1994–2016) showed a decrease in the consumption of various substances. Data from the historical series of the survey show, for example, that the consumption of alcohol (always in the previous 30 days) dropped from 75.1% to 67% among Spanish teenagers between 1994 and 2016, while the consumption of cigarettes decreased from 31.1% to 23%. In the same period, the consumption of cannabis dropped from its historical maximum of 25.1% in 1994 to 13.1% in 2016. On the other hand, the use of other substances that, in the past, were consumed less, such as cocaine, has increased since 2010. For example, according to the last two surveys (that were conducted in 2014 and 2016), the prevalence of consumption of the following illegal substances increased slightly during those years: hypnosedatives (from 10.8% to 11.6%); new substances (from 2.8% to 3.1%); ecstasy (from 0.9% to 1.6%); amphetamines (from 0.9% to 1.2%); methamphetamine (from 0.5% to 1.0%); and, spice (from 0.6% to 0.7%). It is also important to note that the consumption of all illegal drugs is more widespread among men than it is among women, whereas the consumption of legal drugs, such as tobacco, alcohol, and hypnosedatives (with or without prescription), is more widespread among women.

Some studies report that alcohol consumption by female adolescents in Spain significantly increased between 2006 and 2014 and that marijuana and alcohol consumption by friends were associated factors in this increase. Moreover, alcohol consumption increased with age and was more frequent at weekends than on school days. The variables that were associated with a greater probability of alcohol consumption were tobacco, marijuana (OR = 2.37; 95% CI: 2.08–2.72), and alcohol consumption (OR = 7.24; 95% CI: 6.42–8.16) by friends [1].

That the consumption of alcohol, cigarettes and cannabis has not decreased still further might be due to the persistently low perception of their risk to people’s health (especially in regard to alcohol [2]), an issue that is related to perceived invulnerability [3,4,5,6]. The information paradox suggests that information might be less important than one believes when it comes to making decisions [7]. This could explain why Spanish adolescents who perceive themselves to be better informed have a higher prevalence of consuming these substances than those who believe that they are badly informed or only partly informed. 

Methods for tackling and preventing this problem of substance use have mainly focused on transmitting information from expert systems to younger members of the population. Some authors [8] distinguish between prescriptive and participatory models when it comes to introducing public policies that are aimed at preventing consumption. According to Romaní, the predominant model is the prescriptive one, even though it does not enable young people to participate very much in the design of such policies. However, other authors [9] stress the importance of allowing young people to become actively involved in discussions of the advantages and disadvantages of drug use (a ‘pros and cons’ exercise) and negotiate specific and realistic goals for reducing or eliminating their drug consumption. A review of 33 studies published between January 2010 and December 2014 identified several categories of intervention and prevention programs. Only four of these studies took into account a profile of their subjects before implementing their prevention programs. The review concluded that consumption patterns among adolescents should be taken into account when addressing primary prevention [10].

There is evidence that no major transition towards more evidence-based interventions in prevention strategies has taken place in Europe [11]. The predominant approach in Spanish schools is universal prevention that is based on the provision of information. This approach includes specific days on which substance use is discussed and that involve the participation of experts, such as social workers, health professionals, and the police. It means that more systematized programs that incorporate a study plan are secondary, as are prevention programs that are aimed at groups at risk of consumption or that focus on so-called Brief Interventions [12] for users with mild-to-moderate consumption problems [13]. 

In Bukhart’s study of environmental prevention, it was found that adolescents begin to consume these substances influenced by environmental factors, mainly peer-group pressure and reference models [11]. Therefore, alcohol abuse in adolescents might be conditioned by family structure and the role of the adolescent’s father or mother. Specifically, perceived family affection acts as a protective factor against both alcohol consumption and peer group pressure for such consumption [14] and it provides a shared image between parents and adolescents regarding how alcohol should be consumed [15].

This might partly explain the higher consumption of these substances among adolescents who believe they are perfectly or sufficiently informed.

Some studies have shown how predisposition to consumption is stronger among young people, whose reference models are dependent consumers [16]. It has also been shown that the prevalence of alcohol and tobacco consumption is greater among adolescents whose friends have favorable attitudes to the consumption of those substances [17]. Other studies associate a higher consumption of alcohol among adolescents whose fathers, siblings, and especially their friends or best friends often drink [18,19,20]. The literature shows strong concordance between the poly-consumption (the consumption of several substances) of young people and that of their close references, i.e., if an adolescent’s reference is a poly-consumer, it is more likely that he or she will also be one [21]. Some studies stress the need to continue investigating attachment, especially among equals, as a risk and/or protective factor against drug use in adolescence [22]. With regard to poly-consumption, some authors believe that future investigations and interventions should consider social support as a vulnerability marker for the detrimental consequences of substance use and the risk of consumption disorders [23]. Some studies show that, among young Spanish adults (15–34 years old), the factors that are associated with the use of CBP (cannabis and other cannabis-based products), together with the abuse of TSSp (tranquilizers, sedatives, and sleeping pills), were a lack of education (OR 2.34), the consumption of alcohol (OR 7.2), tobacco (OR 6.3) and other illicit psychoactive drugs (OR 6.5), perceived non-health risk for the consumption of CBP and TSSp (OR 3.27), and perceived availability of CBP (OR 2.96) [24].

In this paper, we take these considerations into account to determine whether significant differences exist in the prevalence of consumption among adolescents who report being better or worse informed regarding their consumption of these substances.

## 2. Methods

### 2.1. Design

This paper studies the extent to which teenagers perceive they are informed about substance use and the prevalence of substance consumption. This non-experimental study is descriptive, correlational, and cross-disciplinary, and it is based on a probabilistic sample of students in Spain aged between 14 and 18 (from the ESTUDES survey). Our main objectives are to describe the frequency and distribution of adolescent substance use and establish the possible relationship between the prevalence of consumption of various substances and the determinants of substance use (e.g., sex, age, perceived level of information, consumption by friends, etc.) in an adolescent population. The first sample units were secondary schools. The final observation units were students aged between 14 and 18 who were present at the time that the survey was conducted. The exclusion criteria were student ages over 18 and under 14. The field work was conducted throughout Spain between November 2016 and March 2017. Our main question was whether a relationship exists between being better or more informed regarding substance use and the prevalence of consumption. Our initial hypothesis was that adolescents’ self-perception regarding whether they are informed about substance use does not necessarily imply that their prevalence of consumption is lower. Confirming this hypothesis could open up new lines of research into, for example, the role that is played by the adolescents’ environment in their substance consumption (especially whether their friends are consumers). 

### 2.2. The Sample

This quantitative cross-sectional study used data from the Survey on Drug Use in Spanish Secondary Education (ESTUDES 2016), which was conducted biannually throughout Spain, starting in 1994, as part of the National Plan on Drugs that was developed by the Spanish government’s Ministry of Health, Social Services and Equality. The Survey was funded and promoted by the Spanish Government Delegation for the National Drugs Plan (DGPNSD). Collaborating on the Survey were the governments of the Spanish autonomous communities (Autonomous Community Plans on Drugs and Departments of Education) and the Spanish Ministry of Education, Culture, and Sport. Consent to participate was established in two phases. First, the Departments of Education of the Spanish autonomous communities asked schools to participate in the survey. The schools that were selected then sent a letter to the parents or legal guardians of the adolescents asking for their permission to participate in the survey. This letter explained the survey’s objectives and schedule and stressed the absolute anonymity of the data collection and treatment procedures. Neither the schools nor the promoters of the survey were permitted to know who responded to the survey. After the schools had been selected, survey administrators were sent by the Spanish Observatory on Drugs and Addictions (OEDA) of the Government Delegation for the National Drugs Plan (DGPNSD) to meet with the schools’ principals to randomly select which classes would participate in the survey and establish how to collect the information. All students in the chosen classes completed a standardized, anonymous, and ‘self-administered’ questionnaire while using paper and pencil during their normal class time (45–60 min.). The students’ normal classroom teachers were advised not to be present during the survey so as not to compromise the students’ confidence in the anonymity of their responses. If the teachers were present, they were asked not to walk around the room, explain any survey content, or address the students while they were completing their questionnaire. OEDA survey administrators also ensured that the questionnaires were individually completed. After completing their questionnaires, the students inserted them into blank envelopes, which were then collected by the OEDA staff. 

The universe comprised students that were aged between 14 and 18 who were enrolled at schools or colleges. Sampling was conducted by cluster in two stages: in the first stage, 863 schools were randomly selected; in the second stage, 1726 classes were randomly selected from these schools and the questionnaire was administered to all students in those classes. The final sample comprised 35,370 teenagers. The maximum sample error for a confidence level of 95.5% and *p* = *q* = 0.5 was 0.5% for these Spanish 14–18-year-olds. The sample was weighted by the Spanish autonomous community, the public or private nature of the school, and the type of studies that they offered in order to adapt the proportionality of the sample to the universe (Table 1).

From the initial selection of schools, 91.4% participated in the survey. The remaining 8.6% were replaced, mainly because they declined to collaborate or because they had a high percentage of students aged over 18. The number students present in the classrooms when the survey was administered was 36,371. In the various phases of the process, 1002 questionnaires had to be excluded, either because the respondents’ ages were not appropriate for the study or because the answers were either blank or lacked seriousness. Finally, the total number of questionnaires that were valid for the study was 35,370. The student response rate was 99.78% from the number of questionnaires that were collected and the number excluded for the reasons we have identified. 

### 2.3. Analysis Procedures 

Our statistical model for answering the main question and testing our hypothesis was as follows. At a first descriptive stage, we calculated the prevalence of adolescent substance use by age and sex and the percentages of adolescents who felt that they were well or poorly informed regarding substance use (see the Variables section for an explanation of the question to which the adolescents responded). Additionally, at this stage, we analyzed the adolescents’ responses to a question on the consequences of consumption in several situations. At the second stage of analysis, we used a chi-square test (in which age and sex were controlled) to calculate the possible associations between the prevalence of consumption and whether the adolescents felt they were well or poorly informed. At the third stage, we formulated logistic regression models to calculate the probabilities of consumption by considering variables, such as information and consumption in the adolescents’ environment (i.e., their father, mother, and friends).

Chi-square tests (χ2) were conducted to determine the associations between, on the one hand, the consumption of alcohol (including the getting drunk and binge drinking aspects [25]), cigarettes, and hashish/cannabis, and, on the other, the subject’s perception of the information that they had available on drug use, their gender and age, the consumption of these substances by their mothers, fathers and friends, and their perception of the effects of their consumption. At this level of analysis, it was interesting to observe statistically significant relationships (*p* < 0.05) between categorical variables, particularly if the prevalence of consumption showed some kind of pattern regarding the perception of being better/worst informed, and if this relationship was maintained once sex and age were controlled for. For age and consumption, we used the comparison of means (*t*-test) for independent groups, while we used Cohen’s *d* for effect sizes [26,27,28]. 

Finally, we constructed separate logistic regression models for alcohol consumption, getting drunk, binge drinking (three models), tobacco consumption, and cannabis consumption (two models). The logistic models for alcohol consumption included the variables Mother regularly drinks, Father regularly drinks, some friends drink, and Most friends drink, as well as the variable on whether the adolescents perceived themselves to be well or poorly informed. The logistic models on tobacco and cannabis consumption also included the predictive variables Mother smokes (tobacco), Father smokes (tobacco), Some friends smoke (tobacco), and Most friends smoke (tobacco), as well as the variable on whether the adolescents perceived themselves to be well or poorly informed. 

#### Variables Used

The questionnaire includes a self-perception variable on how well-informed adolescents perceive themselves to be regarding substance use. This question is as follows: Do you feel sufficiently informed about the drugs issue? There were four response categories for this question: Yes, perfectly; Yes, sufficiently; Only partly; and, No, poorly. For the purposes of this article, the categories “Yes, perfectly” and “Yes, sufficiently” were combined to produce the response I believe I am sufficiently or perfectly informed. Similarly, the categories Only partly and No, poorly were combined to produce the response I believe I am partly or badly informed. In this way, the variable on the adolescents’ perception regarding how well-informed they are about substance consumption became a dichotomous variable.

As well as this self-perception variable on how well-informed they feel about substance use, the adolescents were asked to respond to the following: “We would now like to know your opinion regarding the problems (health-related or otherwise) involved in engaging in the following behaviors: (a) smoking a packet of cigarettes every day; (b) smoking 1–5 cigarettes a day; (c) smoking electronic cigarettes; (d) drinking five or six beers or other alcoholic beverages at weekends; (e) drinking one or two beers or other alcoholic beverages every day; (f) drinking five or six beers or other alcoholic beverages every day; (g) taking tranquilizers/sedatives or sleeping pills habitually; (h) smoking hashish or marijuana (cannabis) occasionally; (i) smoking hashish or marijuana (cannabis) habitually; (j) taking powdered cocaine occasionally; (k) taking powdered cocaine habitually; (l) smoking base cocaine/crack occasionally; (m) taking ecstasy occasionally; (n) taking ecstasy habitually; (o) taking amphetamines or speed occasionally; (p) taking amphetamines or speed habitually; (q) taking hallucinogens (LSD, tabs or magic mushrooms) occasionally; (r) taking hallucinogens (LSD, tabs or magic mushrooms) habitually; (s) taking heroin occasionally; (t) taking heroin habitually; (u) injecting drugs occasionally; (v) taking GHB occasionally; (w) taking methamphetamine occasionally; (x) consuming magic mushrooms occasionally; and, (y) taking anabolic steroids occasionally. The response categories for all these consumption situations were: Few or no problems, A lot or quite a lot of problems, and Don’t know. In the results section, we graphically display the percentage of adolescents whose responses were Few or no problems and Don’t know to demonstrate the level of ignorance among adolescents with regard to substance use.

Closed (yes/no) questions were also included to determine whether in the previous 30 days the adolescents: (a) had drunk any type of alcoholic beverage; (b) had smoked tobacco; and, (c) had used cannabis. These were all dichotomous variables. The adolescents were also asked whether in the previous 30 days they had engaged in modes of alcohol consumption, such as: (a) getting drunk and (b) binge drinking. Again, these were both dichotomous variables. 

Finally, included as dependent variables in several logistic regression models were cigarette, alcohol, and cannabis consumption and forms of consumption (getting drunk and binge drinking) (Table 2). For these analyses, we used Wald’s test and present the values of Exp (*β*) and their 95% confidence intervals, CI-Exp (*β*).

In all these analyses, we tested whether gender and age play an important role in the prevalence of consumption, the perception the adolescents have about the level of information available, and the effects and problems that are associated with the consumption of these substances.

In these analyses, we used the SPSS v. 2018 statistical package. The effect sizes were calculated while using the approach outlined on the Psychometrica website [29].

## 3. Results

### 3.1. Adolescents’ Perception of Substance Use

In response to the question Do you feel sufficiently informed about the drugs issue? in the 2016 ESTUDES questionnaire, 66.7% of the adolescents replied that they were sufficiently or perfectly informed and 27% said that they were poorly or only partly informed, while 6.3% did not answer. Teenage girls believed that they were less and/or worse informed (33.7%) than teenage boys (23.9%) (χ2 = 399.8; *p* < 0.000; the effect size was low, i.e., *d* = 0.214). This difference increases with age: at 18 years the percentage points difference is 18 in favor of teenage boys (who believe they are well informed), while, at 14 years, this difference is nine percentage points (also in favor of teenage boys). For the sample as a whole, age 14 is when the lack of information is perceived to be greatest (by 31.5% of all adolescents). 

With regard to the age at which adolescents begin to consume these substances, 27.2% of adolescents who smoke smoked their first cigarette at the age of 14 and 22.9% did so at the age of 15, while 31.4% of those who drink drank their first alcoholic beverage at 14 and 23.6% did so at 15. We observe that a higher percentage of girls (52.1%) than boys (47.7%) started smoking between the ages of 14 and 15 when the data are disaggregated by sex. Similarly, a higher percentage of girls (56.7%) than boys (53.3%) had their first alcoholic drink between the ages of 14 and 15. 

With regard to adolescents who have been drunk, 27.3% got drunk for the first time at the age of 14, while 31.1% did so at the age of 15. Teenage boys and teenage girls did not present significant differences, even in the age at which they began to drink alcoholic beverages practically every week. 

Figure 1 shows the percentages of adolescents who stated that consumption implied few or no problems. The points indicate the percentage of adolescents who did not know how to reply to this question and the dashed line indicates the percentage of adolescents who reported that they were not sufficiently informed.

A proportion of adolescents believed that low levels of consumption have no negative consequences on their health. For example, 27.5% believed that consuming 1–5 cigarettes a day implied hardly any problems, while the figures for drinking five or six beers or other alcoholic beverages at weekends, drinking one or two beers or other alcoholic beverages every day, smoking hashish or marijuana occasionally, and smoking electronic cigarettes were 36.2%, 39%, 37.3%, and 44.6%, respectively.

Adolescents are more aware of the dangers involved in consuming less common substances, such as cocaine, ecstasy, heroin, and amphetamines, etc. Nevertheless, the percentages of non-responses due to ignorance about whether consuming such substances can be harmful to one’s health are high and a proportion of adolescents believes that occasionally consuming them has hardly any consequences on health at all. For cocaine, this figure is 17.2%, while those for ecstasy, amphetamines/speed, hallucinogens, heroin, magic mushrooms, anabolics, and methamphetamines are 14.6%, 14.1%, 14.3%, 13.3%, 12.2%, 10.7%, and 10.1%, respectively. 

Teenage boys, more than teenage girls, underestimate the consequences of consuming the most prevalent substances (cigarettes, alcohol, hashish/marijuana, and electronic cigarettes), especially when their consumption is habitual. For example, 7.8% of teenage boys, as opposed to 6.8% of teenage girls, believe that smoking a packet of cigarettes every day causes few or no problems. The figures for other substances are as follows: smoking electronic cigarettes, 45.1% (for boys) as opposed to 44.1% (for girls); drinking one or two beers or alcohol beverages every day, 41.3% as opposed to 36.7%; drinking five or six beers or alcohol beverages every day, 9.2% as opposed to 7.2%; smoking hashish/marijuana every day, 9.8% as opposed to 6.0%; taking powdered cocaine powder habitually, 3.4% as opposed to 2.5%; regularly taking ecstasy, 3.5% as opposed to 2.5%; taking amphetamines/speed habitually, 3.6% as opposed to 2.4%; and, taking hallucinogens habitually, 3.4% as opposed to 2.4%. In all cases, the differences between teenage boys and teenage girls are statistically significant (*p* < 0.001).

### 3.2. The Consumption of Tobacco and Hashish/Marijuana 

The logistic regression model for the consumption (Yes/No) of tobacco in the previous 30 days enables correct estimation in 82.0% of cases (χ2 = 8375.4; *p* < 0.001), with Nagelkerke R^2^ estimating an adjustment value of 0.356. The probability that an adolescent had smoked tobacco in the previous 30 days when either their mother or father (or both) are smokers was 1.30 times higher than for adolescents whose parents do not smoke (CI = 1.21–1.39; *d* = 0.145). For adolescents, most of whose friends smoke, the probability was 94.0 times higher (CI = 79.3–111.1; *d* = 2.504), while if only some of their friends smoke, the probability was 14.2 times higher (CI = 12.04–16.82). The odds ratio corresponding to the level of information shows that the probability that an adolescent will have smoked tobacco when they believe they are well informed is 1.30 times higher than when they believe that they are not so well informed (CI = 1.21–1.39). 

In the model for hashish/marijuana consumption (Table 3), estimation is correct in 88.6% of cases (χ2 = 6937.4; *p* < 0.001; Nagelkerke R^2^ = 0.370). The probability that an adolescent will have consumed hashish/marijuana in the previous 30 days increases when their mother is a smoker (OR = 1.29; CI = 1.19–1.40: *d* = 0.1404) or their father is a smoker (OR = 1.15; CI = 1.05–1.24; *d* = 0.077), and especially when some of their friends consume hashish/marijuana (OR = 16.32; CI = 14.46–18.41; *d* = 1.539) or most of their friends do (OR = 93.05; CI = 80.55–107.5; *d* = 2.499). The probability that those who believe they are sufficiently informed will have consumed tobacco or hashish/marijuana is 1.59 greater (CI = 1.45–1.75; *d* = 0.2557), in each case, than for those who believe that they are not so well informed. 

We observe that the prevalence relationship is stronger if the majority of the adolescent’s friends smoke in the group who feel better informed (OR = 54.44; CI = 43.11–68.74) than in the group who feel worse informed (OR = 40.10; CI = 27.47–58.55) when consumption is controlled for the variable sufficiently or perfectly informed vs. partly or badly informed. However, there are no significant gender differences in either of these two groups. 

### 3.3. Consumption of Alcohol and Forms of Alcohol Consumption

Estimation is correct in 78.6% of cases (χ2 = 5021.9; *p* < 0.001; Nagelkerke R^2^ = 0.23) in the model for alcohol consumption. The probability that an adolescent will have drunk is 37.3 times higher if most of their friends drink than if they do not (CI = 30.49–45.59; *d* = 1.995). It is also 8.08 times higher when only some of their friends drink (CI = 6.57–9.92; *d* = 1.152). An adolescent’s consumption of alcohol also increases when their mother is a regular drinker (OR = 1.21; CI = 1.08–1.37; *d* = 0.1051) and when they feel better informed (OR = 1.28; CI = 1.19–1.37; *d* = 0.1361). 

In the model for getting drunk (Table 4), estimation is correct in 80.9% of cases (χ^2^ = 6874.5; *p* < 0.01; Nagelkerke R^2^ = 0.303). The probability that an adolescent will have got drunk is 44.8 times higher when most of their friends have got drunk than when they have not (CI = 39.44–50.89; *d* = 2.096). Whether an adolescent gets drunk also depends on whether their mother is a regular drinker (OR = 1.17; CI = 1.03–1.32; *d* = 0.086), whether only some of their friends have got drunk (OR = 8.97; CI = 7.98–10.15; *d* = 1.209), and whether they feel well informed (OR = 1.23; CI = 1.14–1.31; *d* = 0.1141).

Regarding binge drinking, the model provides correct estimation in 80.5% of cases (χ2 = 2912.0; *p* < 0.001; Nagelkerke R^2^ = 0.136). The probability that an adolescent will have binge drunk is 7.36 times higher if most of their friends have binge drunk than if they have not (CI = 6.77–8.00; *d* = 1.1005). This form of consumption depends on whether their fathers drink habitually (OR = 1.11; CI = 1.01–1.20; *d* = 0.0575), whether any of their friends have binge drunk (OR = 2.88; CI = 2.64–3.14; *d* = 0.5832), and whether they perceive themselves to be well informed (OR = 1.23; CI = 1.15–1.31; *d* = 0.1141).

We observe that adolescents who feel worse informed are slightly more exposed to both alcohol consumption and getting drunk when most of their friends have these behaviors than adolescents who feel better informed (unlike what occurred with the consumption of tobacco and hashish/marijuana) when the consumption of alcohol is controlled for the variable sufficiently or perfectly informed vs. badly or partly informed. The OR for alcohol consumption among adolescents when most of their friends drink is 21.4 (CI = 19.46–23.53) in the group who feel better informed and 22.05 (CI = 19.00–25.58) in the group who feel worse informed. When it comes to drunkenness, the OR are 40.34 (CI = 34.94–45–58) and 62.52 (CI = 47.43–82.42), respectively. There were no significant gender differences in either of the two groups or between them, nor were there any differences in the prevalence ratios between those who feel better or worse informed regarding binge drinking. 

## 4. Discussion

Our analysis shows that substance use begins at an early age [30], i.e., before the age of 14, which is in line with other studies [17,31]. Several authors have highlighted the possible consequences of early consumption on habitual consumption and even on risk consumption at a later age [32,33]. Our results show that 31.1% of adolescents smoke their first cigarette before the age of 14 (32% of boys and 30.3% of girls), while 15% try cannabis before the age of 14 (13.7% of girls and 16.2% of boys), 33.1% drink their first alcoholic beverage before the age of 14 (33.3% of boys and 32.8% of girls), and 15.6% get drunk for the first time before the age of 14 (15.4% of boys and 15.8% of girls). 

The percentage of teenage boys and girls who consume these substances increases as they get older. At the age of 14, 11.3% say that they have smoked in the previous 30 days, while at the age of 18 this figure increases to 44.4% (*p* < 0.001; effect size *d* = 0.217). Also, 5% of 14-year-olds and 26.6% of 18-year-olds say they have used cannabis in the previous 30 days (*p* < 0.001; *d* = 0.189), 40.6% of 14-year-olds and 83.3% of 18-year-olds (*p* < 0.001; *d* = 0.318) say they have drunk alcohol, 7.9% of 14-year-olds and 40.4% of 18-year-olds (*p* < 0.001; *d* = 0.259) say they have got drunk, while 9.4% of 14-year-olds and 30% of 18-year-olds (*p* < 0.001; *d* = 0.178) say they have binge drunk.

When consumption (of alcohol, tobacco, or hashish/marijuana) is controlled for the variable sufficiently or perfectly informed vs. badly or partly informed, age and gender no longer have an effect on consumption or forms of consumption. At all ages, adolescents who believe that they are better informed have higher percentages of alcohol, tobacco, and hashish/marijuana consumption than those who believe that they are worse informed. This confirms our hypothesis that adolescents’ self-perception that they are well-informed does not necessarily lead to a lower prevalence of consumption. 

Teenage boys have the same pattern of behavior as teenage girls. Our results suggest that adolescents who believe that they are better informed tend to minimize the risk of effects and problems that are caused by consumption more than those who believe they are less informed and have a greater prevalence for consuming alcohol, tobacco and cannabis. 

The percentage of teenage boys who believe that they are better or more informed is higher than the percentage of teenage girls who do. However, boys tend to underestimate their habitual consumption of substances more than girls, whereas girls underestimate their sporadic consumption more. These data reflect a paradoxical situation in which the best-informed teenage boys underestimate more the damage to their health caused by consuming these substances. Our assertion that adolescents underestimate the effects of substance use is based on the responses that we received to the question “We would now like to know your opinion regarding the problems (health-related or otherwise) involved in engaging in the following behaviors”. Our analyses show that a high percentage of adolescents believe that few or no problems are caused by smoking 1–5 cigarettes a day (27.5%), smoking electronic cigarettes (44.6%), drinking five or six beers or other alcoholic beverages at weekends (36.2%), drinking one or two beers or other alcoholic beverages every day (39%), and smoking hashish/marijuana (cannabis) occasionally (37.3%). In all cases, the adolescents who believe they are “sufficiently or perfectly informed” also have the highest percentages of those who believe that the above situations cause few or no problems. For example, 28.5% of those who believe they are sufficiently or perfectly informed believe that few or no problems are associated with smoking 1 to 5 cigarettes a day, whereas 24.7% of those who think they are not so well informed have that opinion. For the other consumption situations, the figures are as follows: smoking electronic cigarettes, 46.2% vs. 40.6%; drinking five or six beers or other alcoholic beverages at weekends, 37.2% vs. 32.5%; drinking one or two beers or other alcoholic beverages every day, 39.9% vs. 37%; and, using cannabis occasionally, 38.9% vs. 33.5%. In this context, the consumption behaviors of adolescents may be linked to their underestimation of the effects of substance use and we believe, therefore, that a percentage of adolescents underestimates the effects of substance use. 

This suggests that the information adolescents have available might not be of sufficient quality for them to evaluate the negative impact of their consumption. Lack of knowledge or misinformation about the nature of drugs could arouse their curiosity and prevent them from accurately assessing the risks of consumption [34]. 

## 5. Limitations

One of the limitations of this study derives from the question Do you feel sufficiently informed about the drugs issue? because there is no scale and classical methods of reliability, such as calculating Cronbach’s Alpha or the Intraclass Correlation Coefficient (ICC), that can be applied. We admit that no statistical measure can be applied to calculate the reliability and validity of this question, which might raise doubts regarding the authenticity of the responses. However, we could say that there is indirect inter-observer agreement since the responses to this question were similar across the Spanish autonomous communities (inter-observer reliability), as this survey was carried out throughout Spain. This shows that the question was understood and interpreted in the same way. With regard to content validity, we believe that the self-perception indicator does measure what it is supposed to measure, i.e., the extent to which the adolescents feel that they are informed. However, logically, this does not mean that the level to which they are actually informed corresponds to the level at which they perceive themselves to be. This question has been included in the ESTUDES series of surveys, where its capacity to measure self-perception has been qualitatively assessed by researchers and experts.

Additionally, the questionnaire does not inform us of the contents of prevention programs provided to schools by institutions or those of programs implemented by the schools themselves. It would be interesting to know what these programs do and how good the information they provide is in order to understand the statistical association between the adolescents’ high perception of being well-informed and the greater prevalence of consumption among adolescents. 

An issue that is not included in the survey, but that would be interesting to explore, is to determine which channels of information adolescents routinely use to find out about substance use. It would be interesting to analyze the part that is played by the Internet and social networks as information search tools. This analysis could provide clues to enable programs and protocols to be designed for improving the quality of the information that they find. The effect that possessing sufficient quality information would have on the problems and consequences of substance use would justify implementing preventive actions supported by scientific evidence.

A limitation when it comes to comparing these results with those of other studies is that the questionnaire we used does not allow for us to accurately compare our results with those of studies that included the specific item “substance use by my best friend”. Therefore, we have been unable to test the ‘best friend’ hypothesis. Instead, for comparison purposes, we have used data on their friends’ frequency of consumption, which allows for us to roughly determine the association between the adolescents’ prevalence ratios and their friends’ consumption.

## 6. Conclusions

Regardless of their age, teenage boys and teenage girls both who perceive themselves to be well-informed have the highest prevalence of alcohol, cigarette and cannabis consumption in the previous 30 days. They also have the highest prevalence when it comes to modes of alcohol consumption (i.e., getting drunk and binge drinking) [35]. Our results also show that a not-insignificant percentage of adolescents believe that certain consumption situations cause few or no problems and that it is those who perceive themselves to be better informed who believe to a greater extent that such situations cause few or no problems. 

Generally, adolescents with models who are consumers (i.e., father, mother, and/or friends) have a greater prevalence to consume and a lower risk perception [36]. However, the habitual patterns and forms of alcohol and tobacco consumption of the adolescent’s father and mother are not always associated with the patterns and forms of the adolescents themselves. The regression models indicate that, among adolescents, the effect of habitual alcohol consumption is greater when the mother is the habitual consumer and that habitual consumption by the father has little effect. However, this association is reversed when it comes to binge drinking, since, here, the effect is greater when it is the father who is the habitual drinker. With regard to the consumption of tobacco and cannabis, mothers and fathers who smoke are equally associated with the risk that their children will consume these substances [37]. There is a clear relationship between the probability that adolescents will consume alcohol, tobacco, and hashish/marijuana, and whether their friends do. When most friends consume these substances—and, in the case of alcohol, do so by getting drunk and binge drinking—the likelihood that an adolescent will also consume them is much greater than when only some friends or no friends do. Some research suggests that strategies should focus on modifying expectancies, limiting access to alcohol at young ages, and targeting students of higher socioeconomic status to decrease consumption among adolescents in Spain [38].

Future analyses should relate these results to parents’ educational styles. Research suggests that a lenient rather than a heavy-handed educational style might protect against substance use irrespective of the prevailing level of danger in the family’s urban context [39]. This shows the importance of involving families in educational prevention processes based on parental commitment. Similarly, some authors [40] highlight the importance of parental commitment combined with responsible supervision for protecting adolescents against consumption, rather than authoritarian parental styles, which are less effective in providing this protection and have even been identified as risk factors [41]. 

Finally, we should bear in mind that new substances begin to appear over time. Therefore, we need to incorporate new analysis criteria that are more sensitive to consumption patterns, especially the more problematic ones [42]. We also need to progressively incorporate measures pertaining to other drugs that begin to be consumed at a later age. Therefore, prevention must now respond to new challenges, such as: new forms of addiction to, for example, gambling, social networks and the Internet; the social acceptance of substances such as alcohol; the few truly well-informed perceptions among adolescents of the effects of alcohol, tobacco, and cannabis consumption; the pressure to legalize cannabis; and, the emergence of new psychoactive substances onto the market.

## Figures and Tables

**Figure 1 ijerph-17-00627-f001:**
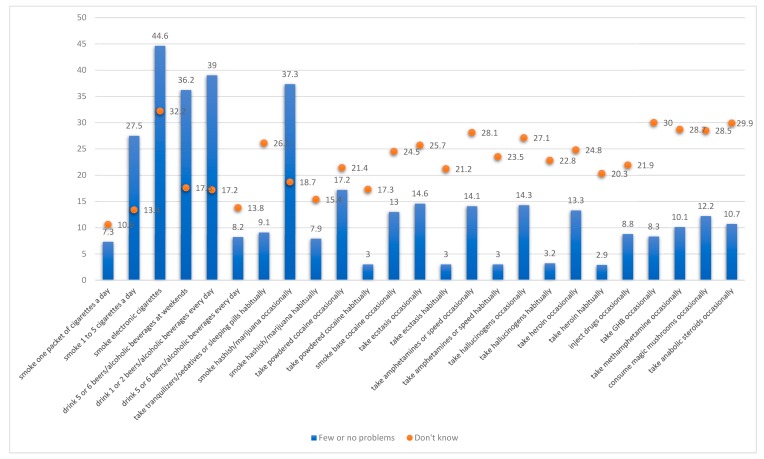
Percentage of adolescents who believe that the consumptions shown pose few or no health problems. Note: This Figure shows the percentage (bars) of adolescents who believe that the consumption situations shown involve few or no problems as well as the percentage (circles) who do not know how to answer or have no opinion on the matter. In each case, the total percentage would reach 100% if we added those adolescents who believe that these situations involve a lot or quite a lot of problems. Source: Authors’ own based on data from ESTUDES 2016.

**Table 1 ijerph-17-00627-t001:** Number (percentage) of subjects in the sample by age and gender.

	14	15	16	17	18	Total
Boys	4535	3783	4660	3615	1288	17,881
	(12.8%)	(10.7%)	(13.2%)	(10.2%)	(3.6%)	(50.6%)
Girls	4412	3555	4713	3685	1124	17,489
	(12.5%)	(10.1%)	(13.3%)	(10.4%)	(3.2%)	(49.4%)
Total	8947	7338	9373	7300	2412	35,370
	(25.3%)	(20.7%)	(26.5%)	(20.6%)	(6.8%)	(100%)

Source: Authors’ own based on data from ESTUDES 2016.

**Table 2 ijerph-17-00627-t002:** Independent variables used in logistic regression analysis.

Age	14–18 Years
Which statement best reflects your mother’s/father’s alcohol consumption in the last 30 days?	0. He/she does not normally drink (Reference Category) 1. He/she drinks regularly
Does your mother or father smoke?	0. No, he/she doesn’t (Ref. Cat.) 1. Yes, he/she does
How many of your friends have smoked tobacco in the last 30 days?	0. None of them (Ref. Cat.) 1. Some of them 2. Most of them
How many of your friends have consumed hashish/marijuana in the last 30 days?	0. None of them (Ref. Cat.) 1. Some of them 2. Most of them
How many of your friends have drunk alcohol in the last 30 days?	0. None of them (Ref. Cat.) 1. Some of them 2. Most of them
How many of your friends have got drunk in the last 30 days?	0. None of them (Ref. Cat.) 1. Some of them 2. Most of them
How many of your friends have binge drunk in the last 30 days?	0. None of them (Ref. Cat.) 1. Some of them 2. Most of them
Do you feel sufficiently informed about the drugs issue?	0. I believe I am partly or badly informed (Ref. Cat.) 1. I believe I am sufficiently or perfectly informed

Source: Authors’ own based on information from ESTUDES 2016.

**Table 3 ijerph-17-00627-t003:** Results of logistic regression analysis for the consumption of tobacco and hashish/marijuana.

	B	E.T.	Wald	*p*	Exp (β)	95% CI
Consumption of tobacco in the previous 30 days						
Mother smokes	0.263	0.034	58.53	0.000	1.30	1.21–1.39
Father smokes	0.265	0.034	61.52	0.000	1.30	1.22–1.39
Some friends smoke	2.656	0.085	970.10	0.000	14.23	12.04–16.82
Most friends smoke	4.544	0.087	2731.41	0.000	94.05	79.31–111.5
Sufficiently or perfectly informed	0.262	0.036	52.50	0.000	1.30	1.21–1.39
Constant	−4.475	0.088	2562.20	0.000	0.01	
Consumption of hashish/marijuana in the previous 30 days						
Mother smokes	0.256	0.042	36.74	0.000	1.29	1.19–1.40
Father smokes	0.136	0.042	10.50	0.000	1.15	1.05–1.24
Some friends smoke	2.792	0.062	2051.42	0.000	16.32	14.46–18.41
Most friends smoke	4.533	0.074	3794.50	0.000	93.05	80.55–107.5
Sufficiently or perfectly informed	0.466	0.048	95.08	0.000	1.59	1.45–1.75
Constant	−4.528	0.071	4097.47	0.000	0.01	

Note: B = Regression coefficient; E.T. = standard error; *p* = probability; Exp (β) = odd ratio; 95% CI = 95% confidence interval. In all cases, the responses Mother smokes, Father smokes, Some friends smoke and Most friends smoke refer only to tobacco consumption. Source: Authors’ own based on data from ESTUDES 2016.

**Table 4 ijerph-17-00627-t004:** Results of logistic regression analysis for alcohol consumption and forms of consumption.

	B	E.T.	Wald	*p*	Exp (β)	95% CI
Consumption of alcohol in the previous 30 days						
Mother drinks habitually	0.194	0.061	10.16	0.000	1.21	1.08–1.37
Father drinks habitually	−0.02	0.044	0.2	0.66	0.98	0.89–1.07
Some friends drink	2.089	0.105	395.49	0.000	8.08	6.57–9.92
Most friends drink	3.619	0.103	1242.48	0.000	37.29	30.49–45.59
Sufficiently or perfectly informed	0.247	0.034	52.65	0.000	1.28	1.19–1.37
Constant	−4.281	0.104	1685.76	0.000	0.01	
Got drunk in the previous 30 days						
Mother drinks habitually	0.157	0.064	6.10	0.010	1.17	1.03–1.32
Father drinks habitually	0.016	0.046	0.12	0.73	1.02	0.93–1.11
Some friends have got drunk	2.194	0.063	1193.76	0.000	8.97	7.98–10.15
Most friends have got drunk	3.802	0.065	3419.43	0.000	44.81	39.44–50.89
Sufficiently or perfectly informed	0.204	0.035	33.30	0.000	1.23	1.14–1.31
Constant	−3.745	0.066	3265.20	0.000	0.024	
Have binge drunk in the previous 30 days						
Mother drinks habitually	0.06	0.062	0.95	0.33	1.06	0.94–1.19
Father drinks habitually	0.1	0.043	5.28	0.02	1.11	1.01–1.20
Some friends have binge drunk	1.058	0.044	582.07	0.000	2.88	2.64–3.14
Most friends have binge drunk	1.996	0.042	2225.82	0.000	7.36	6.77–8.00
Sufficiently or perfectly informed	0.204	0.033	37.52	0.000	1.23	1.15–1.31
Constant	−2.744	0.044	3834.09	0.000	0.06	

Note: B = Regression coefficient; E.T. = standard error; *p* = probability; Exp (β) = odds ratio; 95% CI = 95% confidence interval. Source: Authors’ own based on data from ESTUDES 2016.

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
