# Peer review of "Substance Use among Spanish Adolescents: The Information Paradox"

_ijerph, 2020, doi:10.3390/ijerph17020627_

Round 1

Reviewer 1 Report

Dear authors, it is advisable to add new relevant references on the object of study of the last three years 2017/18 and 19.

Author Response

1. Extensive editing of English language and style required.

The article has been translated and revised by an experienced native English translator.

2. Dear authors, it is advisable to add new relevant references on the object of study of the last three years 2017/18 and 19.

Lines 51-56:

“Some studies report that alcohol consumption by female adolescents in Spain increased significantly between 2006 and 2014 and that marijuana and alcohol consumption by friends were associated factors in this increase. Moreover, alcohol consumption increased with age and was more frequent at weekends than on school days. The variables associated with a greater probability of alcohol consumption were tobacco, marijuana (OR = 2.37; 95% CI: 2.08–2.72) and alcohol consumption (OR = 7.24; 95% CI: 6.42–8.16) by friends [1].”

Lines 72-76:

“A review of 33 studies published between January 2010 and December 2014 identified several categories of intervention and prevention programs. Only four of these studies took into account a profile of their subjects before implementing their prevention programs. The review concluded that consumption patterns among adolescents should be taken into account when addressing primary prevention [10].”

Lines 87-91:

“Alcohol abuse in adolescents may therefore be conditioned by family structure and the role of the adolescent’s father or mother. Specifically, perceived family affection acts as a protective factor against both alcohol consumption and peer group pressure for such consumption [14] and provides a shared image between parents and adolescents regarding how alcohol should be consumed [15].”

We have added new reference:

Alonso-Fernández, N.; Jiménez-Trujillo, I.; Hernández-Barrera, V.; Palacios-Ceña, D.; Carrasco-Garrido, P. Alcohol Consumption Among Spanish Female Adolescents: Related Factors and National Trends 2006–2014. J. Environ. Res. Public Health 2019, 16, 4294. Demant, J.; Schierff, L. M. Five typologies of alcohol and drug prevention programmes. A qualitative review of the content of alcohol and drug prevention programmes targeting adolescents. Drugs: Education, Prevention and Policy 2019, 26(1), 32-39. Olivares, J.; Charro Baena, B.; Prieto Úrsua, M.; Meneses Falcón, C. Estructura familiar y consumo de alcohol en adolescentes. Health & Addictions/Salud y Drogas 2018, 18(1), 107-118. Simonen, J.; Törrönen, J.; Tigerstedt, Ch.; Scheffels, J.; Synnøve Moan, I.; Karlsson, N. Do teenagers’ and parents’ alcohol-related views meet? – Analysing focus group data from Finland and Norway. Drugs: Education, Prevention and Policy 2019, 26(1), 88-96. Tinajero, C.; Cadaveira, F.; Rodríguez, M.S.; Páramo, M.F. Perceived Social Support from Significant Others among Binge Drinking and Polyconsuming Spanish University Students. J. Environ. Res. Public Health 2019, 16, 4506. Palacios-Ceña, D.; Jiménez-Trujillo, I.; Hernández-Barrera, V.; Lima Florencio, L.; Carrasco-Garrido, P. Time Trends in the Co-use of Cannabis and the Misuse of Tranquilizers, Sedatives and Sleeping Pills among Young Adults in Spain between 2009 and 2015. J. Environ. Res. Public Health 2019, 16, 3423. Vargas-Martínez, A. M.; Trapero-Bertran, M.; Gil-García, E.; Lima-Serrano, M. Impact of Binge Drinking (BD) in Adolescence. Are we doing it right? Adicciones 2018, 30(2), 152-155.

Reviewer 2 Report

The study reported has a large and apparently representative sample of adolescents who responded to questions about substance use, as well as questions about parental and social network use.  The presentation could be improved in a number of ways.  While apparently a governmental sponsored long standing survey protocol, the manuscript would be strengthened by describing how consent was obtained, if responses were anonymous, and what the response rate was.  Further, the authors introduce that drugs other and alcohol, tobacco, and marijuana are now more prevalent, but no information is provided about those drugs.  The authors should clarify why.  Finally, there are a number of issues with understanding the results and discussion.  First, the variable of information level is not explained well and no information is proved about the validity or reliability of this variable. How would adolescents interpret a question that asked if they had "perfect" information?  Also throughout the results and discussion there is mention that adolescents 'underestimate' the effects of substance use.  How was this concept defined? One assumes its related to the 'information' variable.  Perhaps giving the exact survey question and scale would be helpful, along with any support for its validity and reliability.

Other issues: Line 114-there seems to be a typo in the sentence explaining Cohen's d.  Line 128 (and other examples) what is meant by the reference source?.  Figure 1 is very confusing. Its not clear how to interpret the 'information' line.  I suggest this be reworked and the dependent variable be clearly defined. Line 177-the variable needs to be clarified around health consequences.  This was not mentioned in the methods. Lines 180-188-Are the differences between sexes statistically significant? Line 207- is smoking tobacco and marijuana being combined? Table 3, is smoking just cigarettes? Line 214, please explain how the analysis by 'information group' was conducted.  Was it a separate group analysis or an interaction term? Line 277, its not clear what participants are 'underestimating'.  While the manuscript and data may have value in understanding adolescent substance use and predictors of such use, the methodological and reporting issues detract from the potential contribution.  In addition, only the variable about "information" is new in the literature, yet its so poorly defined and hard to understand as written, it does not make a strong contribution to the literature.

Author Response

1. English language and style are fine/minor spell check required.

An experienced native English translator has revised the whole text and made any necessary stylistic and typographical amendments.

1. The study reported has a large and apparently representative sample of adolescents who responded to questions about substance use, as well as questions about parental and social network use. 

1.1. The presentation could be improved in a number of ways.  While apparently a governmental sponsored long standing survey protocol, the manuscript would be strengthened by describing how consent was obtained, if responses were anonymous, and what the response rate was. 

We have added the following text on lines 150 to 171:

“The Survey was funded and promoted by the Spanish Government Delegation for the National Drugs Plan (DGPNSD). Collaborating on the Survey were the governments of the Spanish autonomous communities (Autonomous Community Plans on Drugs and Departments of Education) and the Spanish Ministry of Education, Culture and Sport. Consent to participate was established in two phases. First, the Departments of Education of the Spanish autonomous communities asked schools to participate in the survey. The schools that were selected then sent a letter to the parents or legal guardians of the adolescents asking for their permission to participate in the survey. This letter explained the survey’s objectives and schedule and stressed the absolute anonymity of the data collection and treatment procedures. Neither the schools nor the promoters of the survey were permitted to know who responded to the survey. After the schools had been selected, survey administrators were sent by the Spanish Observatory on Drugs and Addictions (OEDA) of the Government Delegation for the National Drugs Plan (DGPNSD) to meet with the schools’ principals to randomly select which classes would participate in the survey and establish how to collect the information. A standardized, anonymous and ‘self-administered’ questionnaire was completed using paper and pencil by all students in the chosen classes during their normal class time (45-60 minutes). The students’ normal classroom teachers were advised not to be present during the survey so as not to compromise the students’ confidence in the anonymity of their responses. If the teachers were present, they were asked not to walk around the room, explain any survey content, or address the students while they were completing their questionnaire. OEDA survey administrators also ensured that the questionnaires were completed individually. After completing their questionnaires, the students inserted them into blank envelopes, which were then collected by the OEDA staff.”

1.2. With regard to the response rate, we have added the following text on lines 185 to 192:

“From the initial selection of schools, 91.4% participated in the survey. The remaining 8.6% were replaced mainly because they declined to collaborate or because they had a high percentage of students aged over 18. The number students present in the classrooms when the survey was administered was 36,371. In the various phases of the process, 1,002 questionnaires had to be excluded either because the respondents’ ages were not appropriate for the study or because the answers were either blank or lacked seriousness. Finally, the total number of questionnaires that were valid for the study was 35,369. From the number of questionnaires collected and the number excluded for the reasons we have identified, the student response rate was 99.78%.”

2. Further, the authors introduce that drugs other and alcohol, tobacco, and marijuana are now more prevalent, but no information is provided about those drugs.  The authors should clarify why.

We have added the following text on lines 43 to 50:

“For example, according to the last two surveys (conducted in 2014 and 2016), the prevalence of consumption of the following illegal substances increased slightly during those years: hypnosedatives (from 10.8% to 11.6%); new substances (from 2.8% to 3.1%); ecstasy (from 0.9% to 1.6%); amphetamines (from 0.9% to 1.2%); methamphetamine (from 0.5% to 1.0%), and spice (from 0.6% to 0.7%). It is also important to note that the consumption of all illegal drugs is more widespread among men than it is among women, whereas the consumption of legal drugs such as tobacco, alcohol and hypnosedatives (with or without prescription) is more widespread among women.

3. Finally, there are a number of issues with understanding the results and discussion. 

3.1. First, the variable of information level is not explained well and no information is proved about the validity or reliability of this variable. How would adolescents interpret a question that asked if they had "perfect" information? Also throughout the results and discussion there is mention that adolescents 'underestimate' the effects of substance use.  How was this concept defined? One assumes its related to the 'information' variable.  Perhaps giving the exact survey question and scale would be helpful, along with any support for its validity and reliability.

On lines 216-249 we have added a subsection, entitled Variables used, to the Methodology section.

In this new subsection we explain how the variables and response categories have been formulated. We have placed special emphasis on explaining the variables: Do you feel sufficiently informed about the drugs issue?; and We would now like to know your opinion regarding the problems (health-related or otherwise) involved in engaging in the following behaviors. The latter variable, which includes numerous consumption situations, measures the extent to which the adolescents perceive themselves to be well informed. The text we have added is as follows:

“The questionnaire includes a self-perception variable on how well-informed adolescents perceive themselves to be regarding substance use. This question is as follows: Do you feel sufficiently informed about the drugs issue? There were four response categories for this question: Yes, perfectly; Yes, sufficiently; Only partly and No, poorly. For the purposes of this article, the categories “Yes, perfectly” and “Yes, sufficiently” were combined to produce the response I believe I am sufficiently or perfectly informed. Similarly, the categories Only partly and No, poorly were combined to produce the response I believe I am partly or badly informed. In this way, the variable on the adolescents’ perception of how well-informed they are about substance consumption became a dichotomous variable.

As well as this self-perception variable on how well-informed they feel about substance use, the adolescents were asked to respond to the following: “We would now like to know your opinion regarding the problems (health-related or otherwise) involved in engaging in the following behaviors: a) smoking a packet of cigarettes every day; b) smoking 1–5 cigarettes a day; c) smoking electronic cigarettes; d) drinking 5 or 6 beers or other alcoholic beverages at weekends; e) drinking 1 or 2 beers or other alcoholic beverages every day; f) drinking 5 or 6 beers or other alcoholic beverages every day; g) taking tranquilizers/sedatives or sleeping pills habitually; h) smoking hashish or marijuana (cannabis) occasionally; i) smoking hashish or marijuana (cannabis) habitually; j) taking powdered cocaine occasionally; k) taking powdered cocaine habitually; l) smoking base cocaine/crack occasionally; m) taking ecstasy occasionally; n) taking ecstasy habitually; o) taking amphetamines or speed occasionally; p) taking amphetamines or speed habitually; q) taking hallucinogens (LSD, tabs or magic mushrooms) occasionally; r) taking hallucinogens (LSD, tabs or magic mushrooms) habitually; s) taking heroin occasionally; t) taking heroin habitually; u) injecting drugs occasionally; v) taking GHB occasionally; w) taking methamphetamine occasionally; x) consuming magic mushrooms occasionally; y) taking anabolic steroids occasionally. The response categories for all these consumption situations were: Few or no problems, A lot or quite a lot of problems and Don’t know. In the results section, we graphically display the percentage of adolescents whose responses were Few or no problems and Don’t know in order to demonstrate the level of ignorance among adolescents with regard to substance use.

Closed (yes/no) questions were also included to determine whether in the previous 30 days the adolescents: a) had drunk any type of alcoholic beverage; b) had smoked tobacco; and c) had used cannabis. These were all dichotomous variables. The adolescents were also asked whether in the previous 30 days they had engaged in modes of alcohol consumption such as: a) getting drunk, and b) binge drinking. Again, these were both dichotomous variables.”   

In the Limitations section, on lines 439-456, we have added the following text in relation to the question about information:

“One of the limitations of this study derives from the question Do you feel sufficiently informed about the drugs issue? because there is no scale and classical methods of reliability such as calculating Cronbach’s Alpha or the Intraclass Correlation Coefficient (ICC) cannot be applied. We admit that no statistical measure can be applied to calculate the reliability and validity of this question, which may raise doubts over the authenticity of the responses. However, as this survey was carried out throughout Spain, we could say that there is indirect inter-observer agreement since the responses to this question were similar across the Spanish autonomous communities (inter-observer reliability). This shows that the question was understood and interpreted in the same way. With regard to content validity, we believe that the self-perception indicator does measure what it is supposed to measure, i.e. the extent to which the adolescents feel they are informed. However, logically, this does not mean that the level to which they are actually informed corresponds to the level at which they perceive themselves to be. This question has been included in the ESTUDES series of surveys, where its capacity to measure self-perception has been qualitatively assessed by researchers and experts.”

To clarify the point about adolescents underestimating the effects of consumption, we have added the following paragraph to the Discussion section on lines 414-431:

“Our assertion that adolescents underestimate the effects of substance use is based on the responses we received to the question “We would now like to know your opinion regarding the problems (health-related or otherwise) involved in engaging in the following behaviors”. Our analyses show that a high percentage of adolescents believe that few or no problems are caused by smoking 1–5 cigarettes a day (27.5%), smoking electronic cigarettes (44.6%), drinking 5 or 6 beers or other alcoholic beverages at weekends (36.2%), drinking 1 or 2 beers or other alcoholic beverages every day (39%) and smoking hashish/marijuana (cannabis) occasionally (37.3%). In all cases, the adolescents who believe they are “sufficiently or perfectly informed” also have the highest percentages of those who believe that the above situations cause few or no problems. For example, 28.5% of those who believe they are sufficiently or perfectly informed believe that few or no problems are associated with smoking 1 to 5 cigarettes a day, whereas 24.7% of those who think they are not so well informed have that opinion. For the other consumption situations, the figures are as follows: smoking electronic cigarettes, 46.2% vs 40.6%; drinking 5 or 6 beers or other alcoholic beverages at weekends, 37.2% vs 32.5%; drinking 1 or 2 beers or other alcoholic beverages every day, 39.9% vs 37%; and using cannabis occasionally, 38.9% vs 33.5%. In this context, the consumption behaviors of adolescents may be linked to their underestimation of the effects of substance use and we believe, therefore, that a percentage of adolescents underestimates the effects of substance use.”

Other issues:

a) Line 114-there seems to be a typo in the sentence explaining Cohen's d. 

The typographical error in relation to Cohen’s d (which appeared on line 114 of the first version of the article) has been corrected.

b) Line 128 (and other examples) what is meant by the reference source?

The source has been corrected in all tables and figures: “Authors’ own based on data from ESTUDES 2016”.

c) Figure 1 is very confusing. Its not clear how to interpret the 'information' line.  I suggest this be reworked and the dependent variable be clearly defined.

We have removed the gray line and left the two response categories corresponding to the question on whether the situations described are considered problematic. There is no dependent variable.

We have added the following footnote to this Figure (lines 289-291).

“This Figure shows the percentage (bars) of adolescents who believe that the consumption situations shown involve few or no problems as well as the percentage (circles) who do not know how to answer or have no opinion on the matter. In each case, the total percentage would reach 100% if we added those adolescents who believe that these situations involve A lot or quite a lot of problems.”

d) Line 177-the variable needs to be clarified around health consequences.  This was not mentioned in the methods.

In the Methodology section we have added a more detailed explanation of the variables and clarified how this aspect has been treated in the questionnaire.

e) Lines 180-188-Are the differences between sexes statistically significant?

All the differences observed between sexes are indeed statistically significant (p < 0.001). We have added the following sentence to the end of the paragraph (lines 319–320):

“In all cases, the differences between teenage boys and teenage girls are statistically significant (p <0.001).”

f) Line 207- is smoking tobacco and marijuana being combined?

The smoking of tobacco and hashish/marijuana are not combined. The sentence is therefore now as follows (lines 338–340):

“The probability that those who believe they are sufficiently informed will have consumed tobacco or hashish/marijuana is 1.59 greater (CI = 1.45–1.75; d = 0.2557), in both cases, than for those who believe they are not so well informed.”

g) Table 3, is smoking just cigarettes?

We have corrected the title of Table 3, which now reads as follows:
“Results of logistic regression analysis for the consumption of tobacco and hashish/marijuana”.

We have added the following text to the footnote to this table:

“In all cases, the responses Mother smokes, Father smokes, Some friends smoke and Most friends smoke refer only to tobacco consumption.”

h) Line 214, please explain how the analysis by 'information group' was conducted.  Was it a separate group analysis or an interaction term?

We analyzed prevalence of consumption in both the well-informed group and the poorly informed group when most of the adolescents’ friends smoke. We therefore have an independent variable (information) with two categories and a dependent variable (prevalence). The variable Most of my friends smoke serves to model the relation but an interaction process is not really involved.

i) Line 277, its not clear what participants are 'underestimating'. 

Here we say:

“However, boys tend to underestimate their habitual consumption of substances more than girls, whereas girls underestimate their sporadic consumption more”.

This means that, although they perceive themselves to be better informed, teenage boys more than teenage girls believe that the various consumption situations cause few or even no problems.

j) While the manuscript and data may have value in understanding adolescent substance use and predictors of such use, the methodological and reporting issues detract from the potential contribution.  In addition, only the variable about "information" is new in the literature, yet its so poorly defined and hard to understand as written, it does not make a strong contribution to the literature.

As we mentioned earlier, in the Methodology section we have incorporated further information about the variables involved in our analysis. We believe this information will help to better understand the definition of our key variable, which is adolescents' self-perception regarding their substance use. We have also now included the adolescents’ assessment of various consumption situations. In our opinion, this is also a novel contribution to the literature.

Reviewer 3 Report

The issue of substance use among Spanish adolescents is a topic of great importance. However, there are several aspects in the paper that need to be addressed and revised:

The Introduction and research background - I would suggest to make a solid and comprehensive literature review on this subject, explain methodology in more details. It is a lack of up-to- date references. An exhaustive and updated analysis of the literature that should lead the authors to the construction of the theoretical framework and to the enunciation of the prepositions, which instead are strangely placed after the description of the case study. After this part, and before the description of the case study, it would be opportune to formulate some research questions and the description of the method to answer them. The mathematical model is generally widely described, but there is no adequate explanation to justify its adoption as a methodological approach. This underlines the main criticality of this paper, which is the lack of a precise methodological design linking the objectives of the study with the empirical results.  The empirical implications of this research and its contribution to understanding the topics are scarce. The results are described clearly and in detail, although it would have been preferable to highlight in detail, a greater correlation between empirical analysis and theoretical framework. Therefore, I suggest to the authors to set the structure of the paper in a more organic way in order to link conceptualization, research hypotheses (and/or prepositions), empirical analysis, discussion of results and conclusions. In addition, section Conclusions is much less discussion than results. There should be more literature based discussion of the results. The potential usefulness/application comments seem underdeveloped.  The authors should state more clearly the limitations of the study. Could the authors come up with some more points?

Therefore, it is recommended:

a more accurate approach in clarifying research variables, diligence in using a research model and in formulating research hypotheses, a more careful handling of the interpretation of statistical data, a very thorough analysis of the variable issue, a serious analysis of the study’s limitations, a more systematic, comprehensive and relevant handling of the scientific discourse altogether.    

Author Response

The issue of substance use among Spanish adolescents is a topic of great importance. However, there are several aspects in the paper that need to be addressed and revised:

1.The Introduction and research background -I would suggest to make a solid and comprehensive literature review on this subject, explain methodology in more details. It is a lack of up-to- date references. An exhaustive and updated analysis of the literature that should lead the authors to the construction of the theoretical framework and to the enunciation of the prepositions, which instead are strangely placed after the description of the case study.

- We have incorporated new and more up-to-date references into the Introduction:

Lines 51-56:

“Some studies report that alcohol consumption by female adolescents in Spain increased significantly between 2006 and 2014 and that marijuana and alcohol consumption by friends were associated factors in this increase. Moreover, alcohol consumption increased with age and was more frequent at weekends than on school days. The variables associated with a greater probability of alcohol consumption were tobacco, marijuana (OR = 2.37; 95% CI: 2.08–2.72) and alcohol consumption (OR = 7.24; 95% CI: 6.42–8.16) by friends [1].”

Lines 72-76:

“A review of 33 studies published between January 2010 and December 2014 identified several categories of intervention and prevention programs. Only four of these studies took into account a profile of their subjects before implementing their prevention programs. The review concluded that consumption patterns among adolescents should be taken into account when addressing primary prevention [10].”

Lines 87-91:

“Alcohol abuse in adolescents may therefore be conditioned by family structure and the role of the adolescent’s father or mother. Specifically, perceived family affection acts as a protective factor against both alcohol consumption and peer group pressure for such consumption [14] and provides a shared image between parents and adolescents regarding how alcohol should be consumed [15].”

Lines 103-111:

“With regard to poly-consumption, some authors believe that future investigations and interventions should consider social support as a vulnerability marker for the detrimental consequences of substance use and the risk of consumption disorders [23]. Some studies show that among young Spanish adults (15–34 years old), the factors associated with the use of CBP (cannabis and other cannabis-based products) together with the abuse of TSSp (tranquilizers, sedatives, and sleeping pills) were lack of education (OR 2.34), the consumption of alcohol (OR 7.2), tobacco (OR 6.3) and other illicit psychoactive drugs (OR 6.5), perceived non-health risk for the consumption of CBP and TSSp (OR 3.27), and perceived availability of CBP (OR 2.96) [24].”

We have incorporated new and more up-to-date references:

Alonso-Fernández, N.; Jiménez-Trujillo, I.; Hernández-Barrera, V.; Palacios-Ceña, D.; Carrasco-Garrido, P. Alcohol Consumption Among Spanish Female Adolescents: Related Factors and National Trends 2006–2014. J. Environ. Res. Public Health 2019, 16, 4294. Demant, J.; Schierff, L. M. Five typologies of alcohol and drug prevention programmes. A qualitative review of the content of alcohol and drug prevention programmes targeting adolescents. Drugs: Education, Prevention and Policy 2019, 26(1), 32-39. Olivares, J.; Charro Baena, B.; Prieto Úrsua, M.; Meneses Falcón, C. Estructura familiar y consumo de alcohol en adolescentes. Health & Addictions/Salud y Drogas 2018, 18(1), 107-118. Simonen, J.; Törrönen, J.; Tigerstedt, Ch.; Scheffels, J.; Synnøve Moan, I.; Karlsson, N. Do teenagers’ and parents’ alcohol-related views meet? – Analysing focus group data from Finland and Norway. Drugs: Education, Prevention and Policy 2019, 26(1), 88-96. Tinajero, C.; Cadaveira, F.; Rodríguez, M.S.; Páramo, M.F. Perceived Social Support from Significant Others among Binge Drinking and Polyconsuming Spanish University Students. J. Environ. Res. Public Health 2019, 16, 4506. Palacios-Ceña, D.; Jiménez-Trujillo, I.; Hernández-Barrera, V.; Lima Florencio, L.; Carrasco-Garrido, P. Time Trends in the Co-use of Cannabis and the Misuse of Tranquilizers, Sedatives and Sleeping Pills among Young Adults in Spain between 2009 and 2015. J. Environ. Res. Public Health 2019, 16, 3423. Vargas-Martínez, A. M.; Trapero-Bertran, M.; Gil-García, E.; Lima-Serrano, M. Impact of Binge Drinking (BD) in Adolescence. Are we doing it right? Adicciones 2018, 30(2), 152-155.

In the Methodology section we better describe the composition of the sample, the selection process for the sample units, and the response rate, etc. (lines 150-171):

"The Survey was funded and promoted by the Spanish Government Delegation for the National Drugs Plan (DGPNSD). Collaborating on the Survey were the governments of the Spanish autonomous communities (Autonomous Community Plans on Drugs and Departments of Education) and the Spanish Ministry of Education, Culture and Sport. Consent to participate was established in two phases. First, the Departments of Education of the Spanish autonomous communities asked schools to participate in the survey. The schools that were selected then sent a letter to the parents or legal guardians of the adolescents asking for their permission to participate in the survey. This letter explained the survey’s objectives and schedule and stressed the absolute anonymity of the data collection and treatment procedures. Neither the schools nor the promoters of the survey were permitted to know who responded to the survey. After the schools had been selected, survey administrators were sent by the Spanish Observatory on Drugs and Addictions (OEDA) of the Government Delegation for the National Drugs Plan (DGPNSD) to meet with the schools’ principals to randomly select which classes would participate in the survey and establish how to collect the information. A standardized, anonymous and ‘self-administered’ questionnaire was completed using paper and pencil by all students in the chosen classes during their normal class time (45-60 minutes). The students’ normal classroom teachers were advised not to be present during the survey so as not to compromise the students’ confidence in the anonymity of their responses. If the teachers were present, they were asked not to walk around the room, explain any survey content, or address the students while they were completing their questionnaire. OEDA survey administrators also ensured that the questionnaires were completed individually. After completing their questionnaires, the students inserted them into blank envelopes, which were then collected by the OEDA staff".

In a new subsection in the Methodology section we have improved our explanation of the variables used in our analysis:

On lines 216-249 we have added a subsection, entitled Variables used, to the Methodology section. In this new subsection we explain how the variables and response categories have been formulated. We have placed special emphasis on explaining the variables: Do you feel sufficiently informed about the drugs issue?; and We would now like to know your opinion regarding the problems (health-related or otherwise) involved in engaging in the following behaviors. The latter variable, which includes numerous consumption situations, measures the extent to which the adolescents perceive themselves to be well informed. The text we have added is as follows:

“The questionnaire includes a self-perception variable on how well-informed adolescents perceive themselves to be regarding substance use. This question is as follows: Do you feel sufficiently informed about the drugs issue? There were four response categories for this question: Yes, perfectly; Yes, sufficiently; Only partly and No, poorly. For the purposes of this article, the categories “Yes, perfectly” and “Yes, sufficiently” were combined to produce the response I believe I am sufficiently or perfectly informed. Similarly, the categories Only partly and No, poorly were combined to produce the response I believe I am partly or badly informed. In this way, the variable on the adolescents’ perception of how well-informed they are about substance consumption became a dichotomous variable.

As well as this self-perception variable on how well-informed they feel about substance use, the adolescents were asked to respond to the following: “We would now like to know your opinion regarding the problems (health-related or otherwise) involved in engaging in the following behaviors: a) smoking a packet of cigarettes every day; b) smoking 1–5 cigarettes a day; c) smoking electronic cigarettes; d) drinking 5 or 6 beers or other alcoholic beverages at weekends; e) drinking 1 or 2 beers or other alcoholic beverages every day; f) drinking 5 or 6 beers or other alcoholic beverages every day; g) taking tranquilizers/sedatives or sleeping pills habitually; h) smoking hashish or marijuana (cannabis) occasionally; i) smoking hashish or marijuana (cannabis) habitually; j) taking powdered cocaine occasionally; k) taking powdered cocaine habitually; l) smoking base cocaine/crack occasionally; m) taking ecstasy occasionally; n) taking ecstasy habitually; o) taking amphetamines or speed occasionally; p) taking amphetamines or speed habitually; q) taking hallucinogens (LSD, tabs or magic mushrooms) occasionally; r) taking hallucinogens (LSD, tabs or magic mushrooms) habitually; s) taking heroin occasionally; t) taking heroin habitually; u) injecting drugs occasionally; v) taking GHB occasionally; w) taking methamphetamine occasionally; x) consuming magic mushrooms occasionally; y) taking anabolic steroids occasionally. The response categories for all these consumption situations were: Few or no problems, A lot or quite a lot of problems and Don’t know. In the results section, we graphically display the percentage of adolescents whose responses were Few or no problems and Don’t know in order to demonstrate the level of ignorance among adolescents with regard to substance use.

Closed (yes/no) questions were also included to determine whether in the previous 30 days the adolescents: a) had drunk any type of alcoholic beverage; b) had smoked tobacco; and c) had used cannabis. These were all dichotomous variables. The adolescents were also asked whether in the previous 30 days they had engaged in modes of alcohol consumption such as: a) getting drunk, and b) binge drinking. Again, these were both dichotomous variables.”

We have explained our methodology better by adding the following paragraph (lines 206–213):

“Finally, we constructed separate logistic regression models for alcohol consumption, getting drunk, binge drinking, tobacco consumption and cannabis consumption. The logistic models for alcohol consumption included the variables Mother regularly drinks, Father regularly drinks, Some friends drink and Most friends drink as well as the variable on whether the adolescents perceived themselves to be well or poorly informed. The logistic models on tobacco and cannabis consumption also included the predictive variables Mother smokes (tobacco), Father smokes (tobacco), Some friends smoke (tobacco) and Most friends smoke (tobacco), as well as the variable on whether the adolescents perceived themselves to be well or poorly informed.”

- After this part, and before the description of the case study, it would be opportune to formulate some research questions and the description of the method to answer them. The mathematical model is generally widely described, but there is no adequate explanation to justify its adoption as a methodological approach. This underlines the main criticality of this paper, which is the lack of a precise methodological design linking the objectives of the study with the empirical results.

We have reordered certain sections and incorporated more information on our methodology, including a description of the variables and the design.

We have added a section on study design separate from the section in which we discuss the sample (lines 117–143):

“Design

This paper studies the extent to which teenagers perceive they are informed about substance use and the prevalence of substance consumption. This non-experimental study is descriptive, correlational and cross-disciplinary and is based on a probabilistic sample of students in Spain aged between 14 and 18 (from the ESTUDES survey). Our main objectives are to describe the frequency and distribution of adolescent substance use and to establish the possible relationship between the prevalence of consumption of various substances and the determinants of substance use (e.g. sex, age, perceived level of information, consumption by friends, etc.) in an adolescent population. The first sample units were secondary schools. The final observation units were students aged between 14 and 18 who were present at the time the survey was conducted. Exclusion criteria were student ages over 18 and under 14. The field work was conducted throughout Spain between November 2016 and March 2017. Our main question was whether a relationship exists between being better or more informed about substance use and the prevalence of consumption. Our initial hypothesis was that adolescents’ self-perception regarding whether they are informed about substance use does not necessarily imply that their prevalence of consumption is lower. Confirming this hypothesis could open up new lines of research into, for example, the role played by the adolescents’ environment in their substance consumption (especially whether their friends are consumers). Our statistical model for answering the main question and testing our hypothesis was as follows. At a first descriptive stage, we calculated the prevalence of adolescent substance use by age and sex and the percentages of adolescents who felt they were well or poorly informed regarding substance use (see the Variables section for an explanation of the question to which the adolescents responded). Also at this stage we analyzed the adolescents’ responses to a question on the consequences of consumption in several situations. At the second stage of analysis, we used a chi-square test (in which age and sex were controlled) to calculate possible associations between the prevalence of consumption and whether the adolescents felt they were well or poorly informed. At the third stage, we formulated logistic regression models to calculate the probabilities of consumption by considering variables such as information and consumption in the adolescents’ environment (i.e. their father, mother and friends).”

- The empirical implications of this research and its contribution to understanding the topics are scarce. The results are described clearly and in detail, although it would have been preferable to highlight in detail, a greater correlation between empirical analysis and theoretical framework. Therefore, I suggest to the authors to set the structure of the paper in a more organic way in order to link conceptualization, research hypotheses (and/or prepositions), empirical analysis, discussion of results and conclusions.

We have separated the Discussion from the Conclusion in order to better clarify the contributions and limits of our analysis.

In the Discussion section we have expanded on some of the study’s other limitations (lines 439–456).

“One of the limitations of this study derives from the question Do you feel sufficiently informed about the drugs issue? because there is no scale and classical methods of reliability such as calculating Cronbach’s Alpha or the Intraclass Correlation Coefficient (ICC) cannot be applied. We admit that no statistical measure can be applied to calculate the reliability and validity of this question, which may raise doubts over the authenticity of the responses. However, as this survey was carried out throughout Spain, we could say that there is indirect inter-observer agreement since the responses to this question were similar across the Spanish autonomous communities (inter-observer reliability). This shows that the question was understood and interpreted in the same way. With regard to content validity, we believe that the self-perception indicator does measure what it is supposed to measure, i.e. the extent to which the adolescents feel they are informed. However, logically, this does not mean that the level to which they are actually informed corresponds to the level at which they perceive themselves to be. This question has been included in the ESTUDES series of surveys, where its capacity to measure self-perception has been qualitatively assessed by researchers and experts.

Also, the questionnaire does not inform us of the contents of prevention programs provided to schools by institutions or those of programs implemented by the schools themselves. It would be interesting to know what these programs do and how good the information they provide is in order to understand the statistical association between the adolescents’ high perception of being well-informed and the greater prevalence of consumption among adolescents.”

-In addition, section Conclusions is much less discussion than results. There should be more literature based discussion of the results. The potential usefulness/application comments seem underdeveloped.  The authors should state more clearly the limitations of the study. Could the authors come up with some more points?

In the Conclusions section we briefly summarize the main results of the study (lines 474–480):

“Regardless of their age, both teenage boys and teenage girls who perceive themselves to be well-informed have the highest prevalence of alcohol, cigarette and cannabis consumption in the previous 30 days. They also have the highest prevalence when it comes to modes of alcohol consumption (i.e. getting drunk and binge drinking) [35]. Our results also show that a not-insignificant percentage of adolescents believe that certain consumption situations cause few or no problems and that it is those who perceive themselves to be better informed who believe to a greater extent that such situations cause few or no problems.”

Therefore, it is recommended:

a) a more accurate approach in clarifying research variables,

We have expanded this information in the Variables subsection of the methodology section.

b) diligence in using a research model and in formulating research hypotheses,

We have added a subsection on the Design of the study that states the main question and the hypothesis we aim to test.

c) a more careful handling of the interpretation of statistical data,

We have conducted a thorough revision of the data and their interpretation.

d) a serious analysis of the study’s limitations,

In the Discussion section we have added new information about certain limitations to the study.

e) a more systematic, comprehensive and relevant handling of the scientific discourse altogether.

We have added new bibliographical references and revised the discourse between the theory and the data.

Round 2

Reviewer 2 Report

The authors have made substantial additions in line with the review requests that have considerably improved the paper.

Author Response

Thank you very much for the feedback. The article has been improved thanks to the contributions you have made.

1. We have renamed analysis procedures such as “Statistical model and analysis procedure” (in the Methodology section).

2. We've changed the next paragraph of place. It was in Design and we put it in Statistical model and analysis procedure (in the Methodology section) (lines 185-195): 

Our statistical model for answering the main question and testing our hypothesis was as follows. At a first descriptive stage, we calculated the prevalence of adolescent substance use by age and sex and the percentages of adolescents who felt they were well or poorly informed regarding substance use (see the Variables section for an explanation of the question to which the adolescents responded). Also at this stage we analyzed the adolescents’ responses to a question on the consequences of consumption in several situations. At the second stage of analysis, we used a chi-square test (in which age and sex were controlled) to calculate possible associations between the prevalence of consumption and whether the adolescents felt they were well or poorly informed. At the third stage, we formulated logistic regression models to calculate the probabilities of consumption by considering variables such as information and consumption in the adolescents’ environment (i.e. their father, mother and friends).

3. We have rewritten and somewhat expanded the information in the statistical model and analysis procedure section (in the Methodology section)(lines 196-205):

Chi-square tests (χ2) were conducted to determine the associations between, on the one hand, the consumption of alcohol (including the getting drunk and binge drinking aspects [25]), cigarettes and hashish/cannabis and, on the other, the subject’s perception of the information they had available on drug use, their gender and age, the consumption of these substances by their mothers, fathers and friends, and their perception of the effects of their consumption. At this level of analysis, it was interesting to observe statistically significant relationships (p<0.05) between categorical variables, particularly if the prevalence of consumption showed some kind of pattern regarding the perception of being better/worst informed, and if this relationship was maintained once sex and age were controlled for. For age and consumption, we used the comparison of means (t-test) for independent groups, while for effect sizes we used Cohen’s d [26,27,28].

4. We have removed the phrase (by repetitive) (Statistical model and analysis procedure subsection, Methodology section):

“For the combined study of information available on drugs and the adolescents’ perception of the problems and effects caused by their habitual and/or sporadic consumption of these substances, we used the chi-square test. In some cases we also used the comparison of means (t-test) for independent groups”.

5. We have introduced these parentheses (line 207):

Finally, we constructed separate logistic regression models for alcohol consumption, getting drunk, binge drinking (3 models)…

Reviewer 3 Report

In revising the manuscript, the authors followed the suggestions provided by reviewers. I recommend the publication after minor revision (regarding general approach).

Author Response

Thank you very much for your valuable contributions. We believe that the article has improved substantially, especially in the part of the methodological explanation. We have made the following minor changes:

1. We have renamed analysis procedures such as “Statistical model and analysis procedure” (in the Methodology section).

2. We've changed the next paragraph of place. It was in Design and we put it in Statistical model and analysis procedure (in the Methodology section) (lines 185-195): 

Our statistical model for answering the main question and testing our hypothesis was as follows. At a first descriptive stage, we calculated the prevalence of adolescent substance use by age and sex and the percentages of adolescents who felt they were well or poorly informed regarding substance use (see the Variables section for an explanation of the question to which the adolescents responded). Also at this stage we analyzed the adolescents’ responses to a question on the consequences of consumption in several situations. At the second stage of analysis, we used a chi-square test (in which age and sex were controlled) to calculate possible associations between the prevalence of consumption and whether the adolescents felt they were well or poorly informed. At the third stage, we formulated logistic regression models to calculate the probabilities of consumption by considering variables such as information and consumption in the adolescents’ environment (i.e. their father, mother and friends).

3. We have rewritten and somewhat expanded the information in the statistical model and analysis procedure section (in the Methodology section)(lines 196-205):

Chi-square tests (χ2) were conducted to determine the associations between, on the one hand, the consumption of alcohol (including the getting drunk and binge drinking aspects [25]), cigarettes and hashish/cannabis and, on the other, the subject’s perception of the information they had available on drug use, their gender and age, the consumption of these substances by their mothers, fathers and friends, and their perception of the effects of their consumption. At this level of analysis, it was interesting to observe statistically significant relationships (p<0.05) between categorical variables, particularly if the prevalence of consumption showed some kind of pattern regarding the perception of being better/worst informed, and if this relationship was maintained once sex and age were controlled for. For age and consumption, we used the comparison of means (t-test) for independent groups, while for effect sizes we used Cohen’s d [26,27,28].

4. We have removed the phrase (by repetitive) (Statistical model and analysis procedure subsection, Methodology section):

“For the combined study of information available on drugs and the adolescents’ perception of the problems and effects caused by their habitual and/or sporadic consumption of these substances, we used the chi-square test. In some cases we also used the comparison of means (t-test) for independent groups”.

5. We have introduced these parentheses (line 207):

Finally, we constructed separate logistic regression models for alcohol consumption, getting drunk, binge drinking (3 models)…